# Bioinformatic Analysis of the Value of Mitophagy and Immune Responses in Corneal Endothelial Dysfunction

**DOI:** 10.3390/cimb47080670

**Published:** 2025-08-19

**Authors:** Ruilin Guo, Chenjia Xu, Yi Yu, Minglu Ma, Xiaojuan Dong, Jing Wu, Chen Ouyang, Jie Ling, Ting Huang

**Affiliations:** 1State Key Laboratory of Ophthalmology, Zhongshan Ophthalmic Center, Sun Yat-sen University, Guangdong Provincial Key Laboratory of Ophthalmology and Visual Science, Guangzhou 510060, China; linlin128@126.com (R.G.); yuyi57@mail2.sysu.edu.cn (Y.Y.); mingluma@126.com (M.M.); qingerdong@126.com (X.D.); wujing@gzzoc.com (J.W.); oyc1991@163.com (C.O.); lingjie2018@163.com (J.L.); 2Ningbo Eye Institute, Ningbo Eye Hospital, Wenzhou Medical University, Ningbo 315040, China; xuchj1189@163.com

**Keywords:** corneal endothelial dysfunction, differentially expressed genes, mitophagy, immune infiltration, biomarker

## Abstract

This study was conducted to elucidate the mitophagy-related differentially expressed genes (MRDEGs) in corneal endothelial dysfunction (CED) and to identify key hub genes that could provide insights into the disease pathogenesis and potential targeted therapies. To achieve this, CED models were established in female SD rats, and RNA sequencing of coronal endothelium samples was conducted to generate a self-testing dataset. Comprehensive bioinformatics analyses were executed, which included the identification of differentially expressed genes (DEGs), GO and KEGG enrichment analyses, GSEA, and GSVA. A protein–protein interaction (PPI) network was constructed to identify highly interconnected hub genes, followed by the construction of ROC curves to validate MRDEGs within the dataset, alongside qRT-PCR assays. Our findings revealed a total of 18,511 DEGs, among which 20 genes were characterized as MRDEGs. Enrichment analyses indicated significant associations with monocyte differentiation and lymphocyte proliferation. Importantly, eight hub genes emerged from the PPI network as promising therapeutic targets. In conclusion, this study underscores the important role of MRDEGs and immune infiltration in CED, laying the groundwork for future investigations into targeted therapies for this disease.

## 1. Introduction

Corneal endothelial dysfunction (CED) is an important category of ocular disorders, causing the loss of corneal transparency and visual acuity [1]. Current treatment primarily involves surgical approaches like penetrating keratoplasty and Descemet membrane endothelial keratoplasty [2]. However, alternative therapies are needed because of the donor corneal materials shortage and transplant complications [3,4]. Exploring the molecular mechanisms underlying CED and identifying potential biomarkers could facilitate more effective diagnostic and therapeutic strategies.

Corneal endothelial cells are prone to mitochondrial DNA damage because of their postmitotic nature. The oxidative balance and mitochondrial dysfunction of the cornea become increasingly apparent under disease conditions [5]. And, the enhancement in mitochondrial performance and antioxidant response has been linked to CED, such as Fuchs endothelial corneal dystrophy [6,7] and other related disorders [8].

The advent of high-throughput sequencing technologies and bioinformatics tools has revolutionized the exploration of differentially expressed genes (DEGs) in various diseases [9]. Notably, the identification of DEGs related to mitophagy may elucidate the cellular processes underlying endothelial cell health and disease progression.

This study aims to use bioinformatics analyses to systematically analyze DEGs associated with mitophagy in CED, identifying hub genes to enhance understanding of CED pathogenesis and guide targeted therapies [8,10].

## 2. Materials and Methods

### 2.1. Data Extraction

In accordance with previously reported methods [11,12], a total of 20 SD rats were used, with 10 animals per CED group and control group. The CED model was established by injecting rat recombinant TNF-α and IFN-γ (0.005 mg/kg, Sino Biological, Inc., Beijing, China) into the anterior chamber of the right eye for three consecutive days. Then, they were photographed by a slit-lamp microscope and then examined via AS-OCT (anterior segment optical coherence tomography) to measure central corneal thickness (CCT).

Furthermore, the corneas were incubated with primary antibodies followed by staining using secondary antibodies conjugated with Alexa Fluor 488 or Alexa Fluor 594 (Invitrogen, Carlsbad, CA, USA). The immunofluorescence images were performed on a Zeiss LSM 880 microscope (Carl Zeiss Meditec, Jena, Germany). Primary antibodies included anti-Na^+^-K^+^-ATPase (Cell Signaling Technology, CST; Danvers, MA, USA) and anti-ZO-1 (Santa Cruz, CA, USA).

Total RNA extracted from rat corneal endothelium underwent NovaSeq 6000 platform sequencing (Illumina, San Diego, CA, USA) as the self-testing CED dataset (Table 1). Then, we normalized the self-testing CED dataset via the DESeq2 package (v4.2.2).

In addition, we used the GeneCards database (https://www.genecards.org/ (accessed on 12 April 2023)) [13] to identify mitophagy-related genes (MRGs). We queried “mitophagy” for “Protein Coding” MRG, identifying 4600 MRGs (Appendix A).

### 2.2. DEGs Related to CED

To identify the potential mechanism of CED, we first used the DESeq2 package to standardize and analyze the self-testing CED dataset to obtain the DEGs between the CED and control groups.

DEGs were identified using cutoff thresholds of |logFC| > 1 and *p* < 0.05. Genes satisfying logFC > 1 with *p* < 0.05 were classified as upregulated, while those with logFC < −1 and *p* < 0.05 were designated downregulated DEGs. The results were displayed in a volcano plot and a heatmap via the package ggplot2 (v4.2.2) and the package pheatmap (v4.2.2). To obtain MRDEGs for CED, we intersected the DEGs with MRGs and plotted a Wayne map.

### 2.3. GO and KEGG Analyses of DEGs

A functional enrichment analysis of MRDEGs was conducted using clusterProfiler (v4.2.2) [14] with a Benjamini–Hochberg (BH) correction, screening GO [15] (functional annotation), and KEGG [16] (pathway/disease database) terms at significance thresholds of *p* adj < 0.05 and FDR q < 0.05.

### 2.4. GSEA and GSVA

GSEA evaluates the contribution of gene sets to phenotypes by assessing their distribution trends in the sorted gene list [17]. A GSEA of all genes in the self-testing CED dataset was performed using clusterProfiler with 2020 seeds, 1000 computations, and gene set size 10–500. Significance thresholds were set at *p* adj < 0.05 and an FDR (q value) < 0.05.

GSVA [18,19] was performed to evaluate pathway enrichment in different samples. We obtained the c2.cp.all.V2022.1.Hs.symbols.gmt gene set and performed GSVA analysis, identifying the top 10 positively and negatively correlated pathways based on logFC with *p* adj < 0.05.

### 2.5. PPI Network Analysis

The PPI network consists of protein interactions in signaling and metabolism [20]. It was constructed from the screened MRDEGs using the STRING database [21] with visualization in Cytoscape (v 3.9.1) [22].

Hub genes (top 10 scores) were mined from this network via five topological algorithms: Maximal Clique Centrality (MCC) [23], Density of Maximum Neighborhood Component (DMNC), Maximum Neighborhood Component (MNC), Edge Percolated Component (EPC), and Degree [24,25]. Their protein structures were subsequently predicted using the AlphaFold (https://www.alphafold.ebi.ac.uk/; accessed on 8 April 2023) website [26].

### 2.6. Construction of the mRNA-miRNA, mRNA-TF, and mRNA-RBP Interaction Networks

miRNAs regulate biological development and evolution through multi-target interactions [27]. Using miRDB [28], we predicted the miRNAs targeting hub genes (target score > 80) and visualized them via Cytoscape software.

Transcription factors (TFs) could control gene expression post-transcriptionally [29]. Using the ChIPBase database [30] (version 3.0) (https://rna.sysu.edu.cn/chipbase/ (accessed on 5 May 2023)), we identified thousands of TF binding motifs and predicted TF–gene regulatory relationships, visualized by Cytoscape.

RNA-binding proteins (RBPs) interacting with the hub genes were predicted using the ENCORI [31] database (https://starbase.sysu.edu.cn/ (accessed on15 May 2023)), with results visualized in Cytoscape.

### 2.7. ROC Curve Analysis

The receiver operating characteristic curve (ROC) [32] is a comprehensive indicator of the sensitivity and specificity of continuous variables. We used the proc package to construct the ROC curve of the MRDEGs from the self-testing CED dataset. We evaluated MRDEGs’ diagnostic accuracy for CED occurrence using the area under the ROC curve (AUC). The AUC value ranges from 0.5 (random discrimination) to 1 (perfect discrimination), with interpretation thresholds: 0.5–0.7 = low accuracy, 0.7–0.9 = moderate accuracy, and >0.9 = high accuracy.

### 2.8. Immune Infiltration Analysis

Using the single-sample GSEA (ssGSEA) via the GSVA package [33], we calculated the enrichment scores representing immune cell infiltration levels across samples. This generated 28 tumor-infiltrating immune cell gene sets, with differences between CED and control groups visualized in boxplots. A Spearman correlation analysis assessed hub gene–immune cell relationships using the self-testing CED dataset, visualized through ggplot2 dot plots. Complementarily, we applied CIBERSORT [34] to matrix data combined with the LM22 signature. Samples with immune enrichment scores > 0 were retained to generate the infiltration abundance matrix, displayed in comparative plots. Correlation results were visualized using the ggplot2 and heatmap packages.

### 2.9. qRT-PCR Validation of the Hub Genes

The relative mRNA expression levels of hub genes were quantified via qRT-PCR. Amplifications were conducted on a LightCycler 384 instrument (Roche Life Sciences, Basel, Switzerland) using SYBR Green real-time PCR mix (Vazyme Biotech, Nanjing, China). Gene-specific primers were synthesized by Sino Biological Inc., Beijing, China.

### 2.10. Statistical Analysis

All statistical analyses were conducted using R software (v4.2.2). Continuous variables are expressed as mean ± SD. Group comparisons employed the following non-parametric approaches: Wilcoxon rank-sum tests for dual group contrasts and Kruskal–Wallis examinations for multi-group analyses (≥3 groups). Categorical variables were evaluated via chi-square or Fisher’s exact tests as appropriate. Unless otherwise indicated, Spearman’s correlation coefficients quantified variable associations, with statistical significance defined at *p* < 0.05.

## 3. Results

### 3.1. TNF-α and IFN-γ Induced CED

Post-model establishment, CED-group corneas exhibited edema and opacification, while the corneas were transparent in the control group. AS-OCT revealed significantly thinner corneas in the control group versus the CED group. The corneal endothelial cells of the control group maintained a regular hexagonal shape while CED samples showed cellular enlargement and reduced density (Figure 1A).

### 3.2. Analysis of DEGs Related to CED

After normalizing the self-testing CED dataset, the expression profile data were standardized, and the expression of the sample data tended to be consistent (Figure 1B–E).

The self-testing CED dataset included 18,511 DEGs. A total of 357 genes met the cutoff criteria, including 287 upregulated genes and 70 downregulated genes (Figure 2A). To identify MRDEGs, we converted the 4600 MRGs into rat MRGs via the R package (v4.2.2) homologene. A total of 4207 rat MRGs were obtained after removing duplicate genes (Appendix A). Then, we interlaced the DEGs with the 4207 rat MRGs, which resulted in a total of 20 MRDEGs (Blk, Cftr, Cd79b, Il10, Il9r, Il24, Il23r, BLK, CFTR, CD79B, Il6, Scube3, P2rx5, Tgm3, Ubd, Ablim3, Ccl12, Cd3d, Cntnap4, Kmo, Pnpla5, Spic, and Ubash3a) (Figure 2B). We analyzed the differential expressions of these 20 MRDEGs in the CED and control groups (Figure 2C).

### 3.3. GO and KEGG Analyses of MRDEGs

The functional analysis of the 20 MRDEGs included GO and KEGG enrichment analysis (Figure 2D–H). Furthermore, we calculated gene-specific z scores from the logFC values derived from the differential analysis between the CED and control groups (Figure 2I–L).

### 3.4. GSEA and GSVA of Data from the Self-Testing CED Dataset

The results of the GSEA analysis revealed that the DEGs were significantly enriched in the following pathways: Tcr signaling, T cell receptor signaling pathway, DNA double-strand break response, and Rhodopsin pathway (Figure 3A–E). We performed a GSVA analysis on all genes between the CED and control groups. The top 10 positively and 10 negatively correlated pathways are shown in Table 2 and Figure 3F,G.

### 3.5. Construction of PPI Networks

A PPI analysis of 20 MRDEGs (Blk, Cftr, Cd79b, Il10, Il9r, Il24, Il23r, Il6, Scube3, P2rx5, Tgm3, Ubd, Ablim3, Ccl12, Cd3d, Cntnap4, Kmo, Pnpla5, Spic, and Ubash3a) was conducted. Only the MRDEGs that had connections with other nodes were retained, and a PPI network composed of 16 MRDEGs was ultimately constructed (Figure 4A). Then, the top 10 DEGs were selected (Figure 4B–F), and their intersection revealed eight hub genes: Blk, Cftr, Cd79b, Il10, Il9r, Il24, Il23r, and Il6 (Figure 4G). The protein results were analyzed using AlphaFold (Figure 4H–O).

### 3.6. mRNA-miRNA, mRNA-TF, and mRNA-RBP Interaction Networks

Using mRNA-miRNA data in the miRDB database, we predicted miRNAs interacting with the eight hub genes, yielding a network comprising seven hub genes and 63 miRNAs (Figure 5A).

The ChIPBase database (version 3.0) identified TFs binding to eight hub genes, forming an interaction network of five hub genes and 24 TFs (Figure 5B).

Additionally, we predicted RBPs interacting with these hub genes using ENCORI, leading to a network of seven TRHGs and 23 RBPs (Figure 5C).

### 3.7. Differential Expression Analysis of MRDEGs

We analyzed the differential expression of eight hub genes and used the Wilcoxon signed-rank test to compare these genes in the CED and control groups (Figure 6A). We then constructed ROC curves for each gene (Figure 6B–H). The results revealed that the expression of the seven hub genes was an accurate predictor of the occurrence of CED (AUC = 1.000).

### 3.8. Immune Characteristic Analysis of MRDEGs in the Self-Testing CED Dataset

To explore the immune characteristics of MRDEGs, we converted the rat genes in the self-testing CED dataset into human genes (Appendix A). Using the ssGSEA, we quantified the infiltration levels of 28 immune cells in the CED and control groups (Figure 7A), identifying 24 immune cell types with significantly differential expression levels (*p* value < 0.05). CD56dim natural killer cells showed significant negative correlations with all 23 other immune cells, which exhibited mutual positive correlations (Figure 7B).

Moreover, correlations between 24 immune cell infiltration signatures and eight MRDEG expression patterns were quantified. This revealed a marked inverse relationship between CD56dim natural killer cell abundance and MRDEG levels, in contrast to the positive associations consistently observed for all other immune cell types (Figure 7C).

The results revealed that there were seven types of immune cells with statistically significant differences between the two groups (*p* value < 0.05), including memory B cells, resting dendritic cells, activated mast cells, resting mast cells, monocytes, CD4 naive, and follicular helper T cells (Figure 7D).

Subsequently, we calculated the correlations among the infiltration abundance of seven types of immune cells. There was a significant negative correlation between the infiltration abundance of activated mast cells and naive CD4+ T cells, and the other five immune cell types were significantly positively correlated with each other (Figure 7E).

We calculated the correlation between the infiltration abundance of seven immune cells and the expression levels of eight MRDEGs (Figure 7F). The results revealed a negative correlation between the expression of naive CD4+ T cells and MRDEGs, but a significant positive correlation between the expression of the other six immune cells and MRDEGs.

### 3.9. qPCR Validation of the Hub Genes

The qPCR validation confirmed significantly elevated mRNA expression of Blk, Cftr, Cd79b, Il10, Il9r, Il24, Il23r, and Il6 in the CED group. These findings verify our bioinformatics findings and demonstrate their research potential (Figure 8).

## 4. Discussion

Due to recent advancements in molecular mechanisms, particularly oxidative stress and mitochondrial dysfunction [35,36], the pathogenesis of CED remains unclear, requiring further research into the molecular mechanisms and potential therapeutic targets.

A previous study described that TNF-α and IFN-γ induce the loss of corneal endothelial cells [37,38]. Moreover, our previous research has shown that TNF-α and IFN-γ can induce CED [11], leading to cornea edema and loss of transparency.

Mitophagy involves the clearance of damaged mitochondria through a process [39,40] and is essential for maintaining cellular health and function in the cornea endothelium [35,41]. The upregulated mitophagy genes may reflect cellular responses to stress conditions, whereas the downregulated genes could indicate impaired mitochondrial function. It is a therapeutic target for retinal diseases and other ocular diseases [42], such as age-related macular degeneration [43].

Besides, the findings highlighted the immune response and cell signaling in the corneal environment, consistent with the results of previous studies [44,45]. The analysis of immune cell infiltration revealed significant differences between the CED and controls. There was an increase in activated CD8+ T cells and regulatory T cells, indicating an altered immune landscape. CD8+ T cells can drive chronic inflammation and regulate regulatory T cells in the ocular mucosal immune system [46]. This response was implicated in the pathogenesis of many ocular disorders with an inflammatory basis, such as Sjögren syndrome [47]. Moreover, ocular resident T cells can be induced to become regulatory T cells within the human peripheral microenvironment [48]. These findings echo those of previous studies that linked the immune response to the pathogenesis of various ocular diseases, such as autoimmune keratitis [49] and corneal allograft rejection [50], suggesting the immune system’s role in disease initiation and progression.

A PPI network analysis revealed eight hub genes, including Blk, Cftr, Cd79b, Il10, Il9r, Il24, Il23r, and Il6. Blk encodes a kinase of the Src family (Blk), which is expressed in all stages of B cell development [51,52]. CD79b is essential for B cell development and function [53]. Mutations in the CFTR gene cause cystic fibrosis, and cystic fibrosis transmembrane conductance regulators play a role in water and ion transport across the corneal endothelium [54]. Recent data suggest that CFTR is an attractive target for corneal edema [55]. Il10, Il9r, Il24, Il23r, and Il6 are important regulators of the immune response [56,57,58,59]. The identification of these hub genes emphasized the intricate interplay of these genes in maintaining corneal endothelial cell function and viability, validated by qRT-PCR.

This study has certain limitations that must be acknowledged. First, the sample size utilized for the differential expression analysis herein may have been insufficient to account for the biological variability inherent in CED. While our qPCR analysis confirmed the significant dysregulation of hub mitophagy genes, the direct visualization of the mitophagic flux in the corneal endothelium, which poses specific technical challenges in this model, represents a valuable avenue for future investigation. Moreover, while our bioinformatics approach provided valuable insights into the molecular mechanisms underlying CED, the absence of extensive clinical validation limits the robustness of our conclusions.

## 5. Conclusions

In conclusion, this study revealed the role of MRDEGs in CED, providing a foundation for future research and therapeutic development. Continued exploration of the immune response and the intricate molecular networks involved in CED will be essential for advancing our understanding and management of this group of diseases.

## Figures and Tables

**Figure 1 cimb-47-00670-f001:**
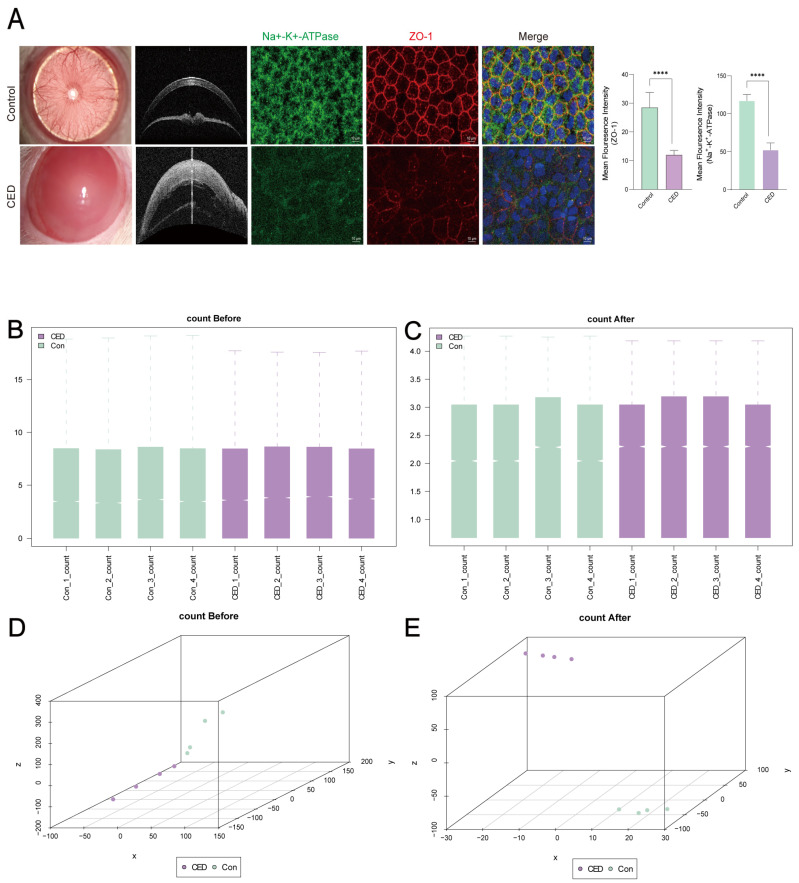
Standardized processing of CED dataset for self-testing dataset. (**A**) Photographs and AS-OCT of rats completed the modeling. Immunofluorescence staining with Na^+^-K^+^-ATPase (green), ZO-1 (red), and DAPI (blue). (**B**,**C**) Boxplot of the self-testing CED dataset before (**B**) and after (**C**) standardization. (**D**,**E**) PCA plots of the self-testing CED dataset before (**D**) and after (**E**) normalization. Data represents mean ± SD. **** *p* value < 0.0001.

**Figure 2 cimb-47-00670-f002:**
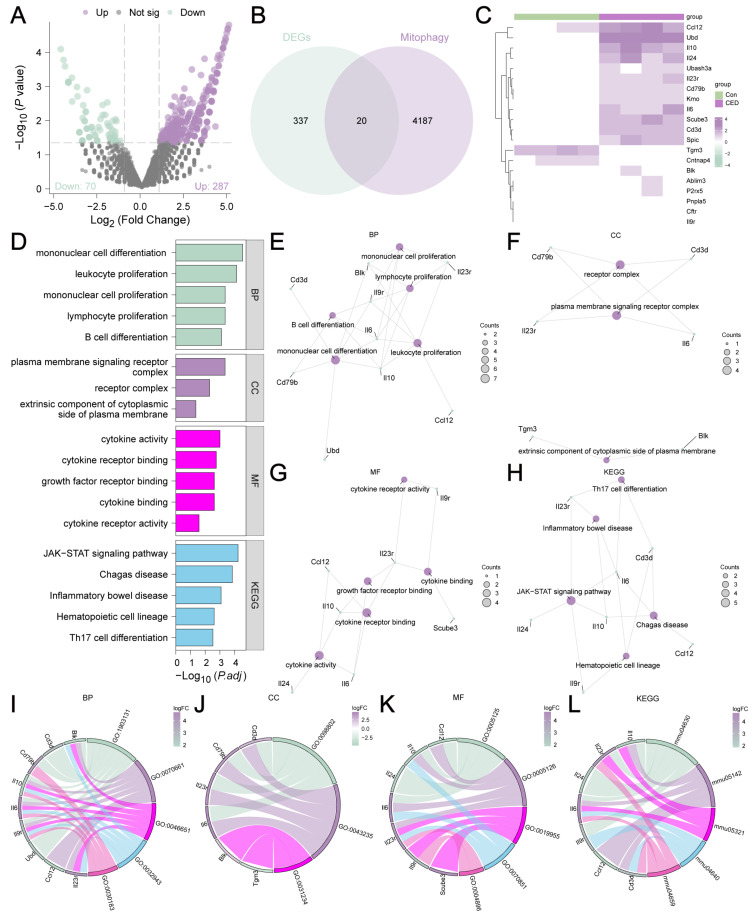
Analysis and enrichment analysis of MRDEGs in the self-testing CED dataset. (**A**) Volcano plot of differential gene analysis of CED and control group. (**B**) Venn diagram of DEGs and MRGs. (**C**) Complex numerical heatmap of MRDEGs. (**D**) Bar chart of GO/KEGG enrichment analysis of MRDEGs. (**E**–**H**) Network diagram of GO/KEGG enrichment analysis. (**I**–**L**) Chordgraph of GO/KEGG enrichment analysis with logFC values of MRDEGs.

**Figure 3 cimb-47-00670-f003:**
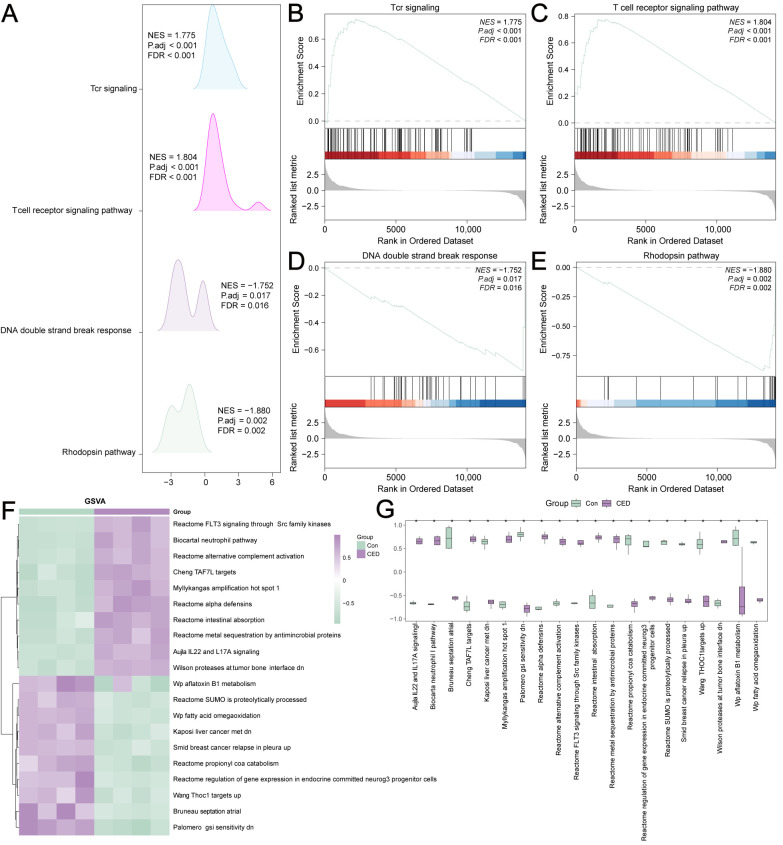
GSEA and GSVA of the self-testing CED dataset. (**A**) Four main biological characteristics of the GSEA. (**B**–**E**) DEGs were enriched in TCR_SIGNALING (**B**), TCELL_RECEPTOR_SIGNALING_PATHWAY (**C**), DNA_DOUBLE_STRAND_BREAK_RESPONSE (**D**), RHODOPSIN_PATHWAY (**E**). (**F**) Heatmap of 20 main biological features of GSVA. (**G**) Group comparison chart of the 20 main biological characteristics of GSVA. Data represents mean ± SD. * *p* value < 0.05.

**Figure 4 cimb-47-00670-f004:**
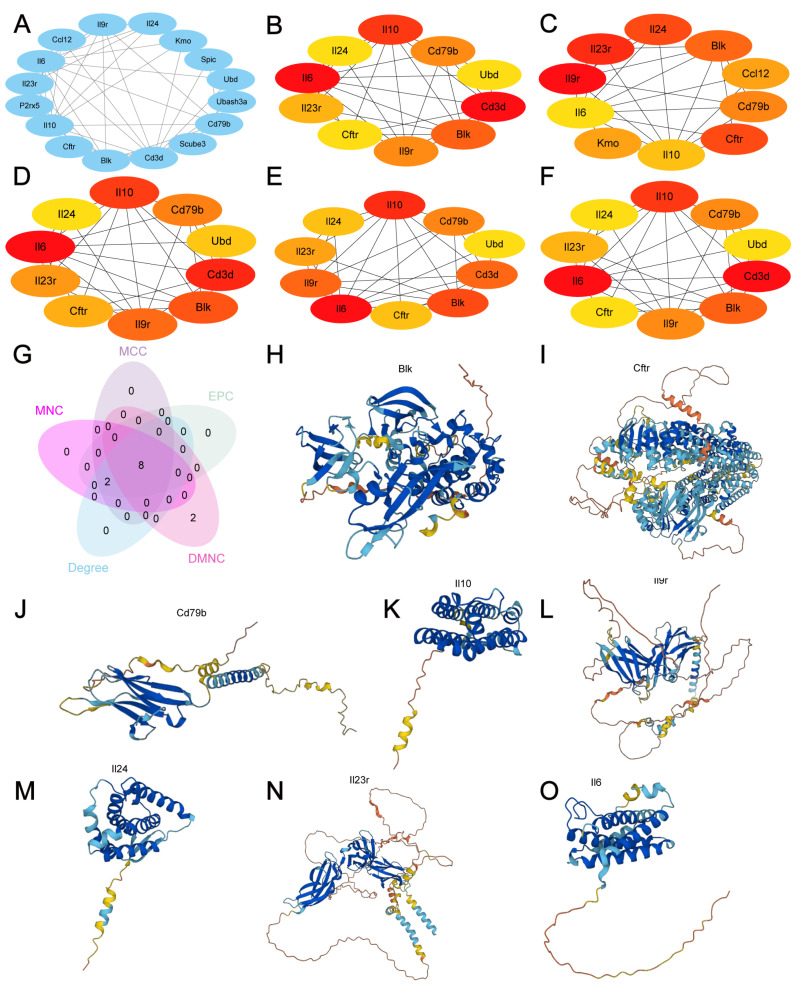
Construction of the PPI network. (**A**) PPI network of MRDEGs. (**B**–**F**) Top 10 node networks based on the Degree (**B**), DMNC (**C**), EPC (**D**), MCC (**E**), and MNC (**F**) algorithms. The color of the rectangular block from yellow to red represents the gradual increase in the score. (**G**) Venn diagram of the top 10 nodes with the protein structures of Blk (**H**), Cftr (**I**), Cd79b (**J**), Il10 (**K**), Il9r (**L**), Il24 (**M**), Il23r (**N**), and Il6 (**O**).

**Figure 5 cimb-47-00670-f005:**
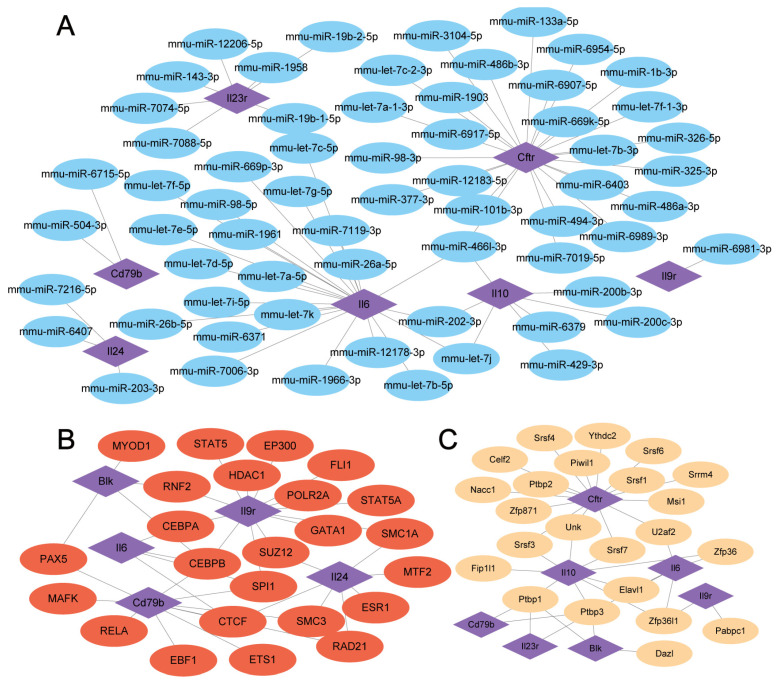
Construction of the mRNA-miRNA, mRNA-TF, and mRNA-RBP interaction networks. (**A**) The blue oval blocks are miRNAs; the purple diamonds are mRNAs. (**B**) The red oval blocks are TFs; the purple diamonds are mRNAs. (**C**) The yellow oval blocks are RBPs; the purple diamonds are mRNAs.

**Figure 6 cimb-47-00670-f006:**
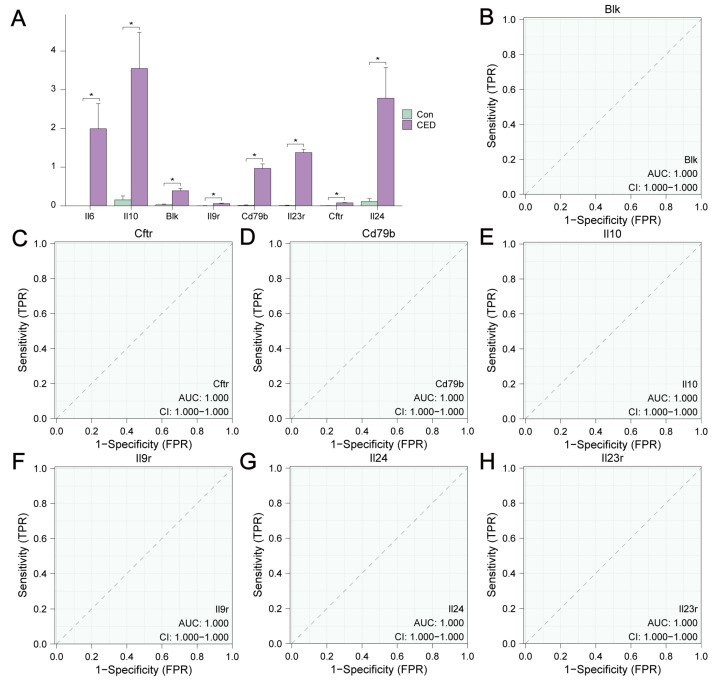
Differential expression analysis of the hub genes in the self-testing CED dataset. (**A**) Group comparison diagram showing the differential expression analysis of 8 hub genes. (**B**–**H**) The ROC curve analysis of 7 hub genes (Blk (**B**), Cftr (**C**), Cd79b (**D**), Il10 (**E**), Il9r (**F**), Il24 (**G**), and Il23r (**H**)). Data represents mean ± SD. * *p* value < 0.05.

**Figure 7 cimb-47-00670-f007:**
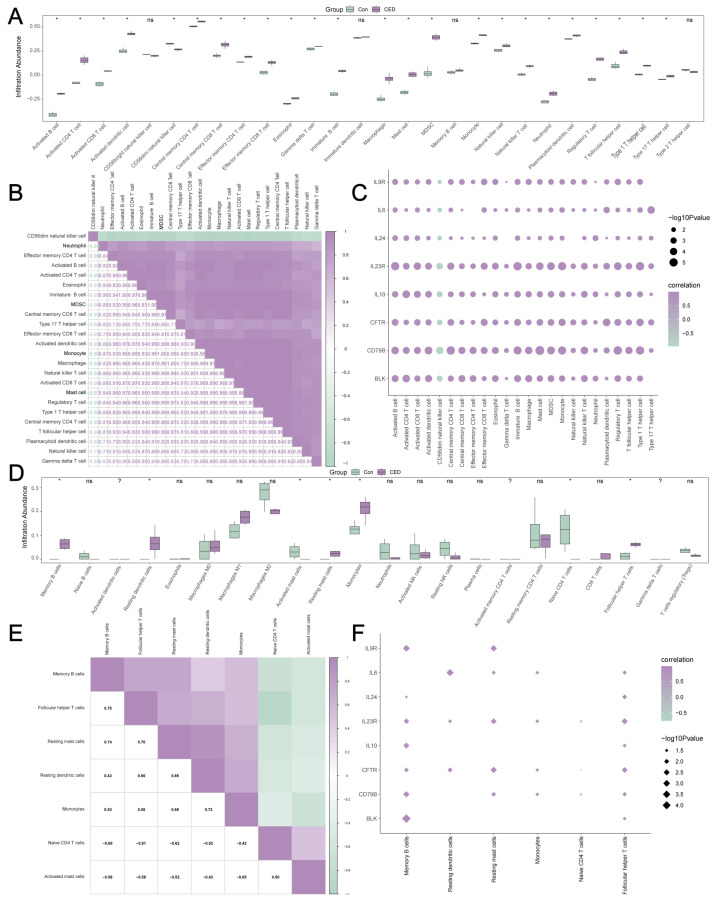
Immune infiltration analysis of the self-testing CED dataset. (**A**) Comparison of the ssGSEA immune infiltration results. (**B**) Correlation analysis of immune cell infiltration abundance. (**C**) Dot plot of the correlations between immune cells and MRDEGs. (**D**) Group comparison diagram of CIBERSORT immune infiltration analysis results. (**E**) Correlation analysis results of immune cell infiltration abundance. (**F**) Dot plot of the correlations between immune cells and MRDEGs. Data represents mean ± SD. ns = no statistical significance. * *p* value < 0.05. ? represented “not detected”.

**Figure 8 cimb-47-00670-f008:**
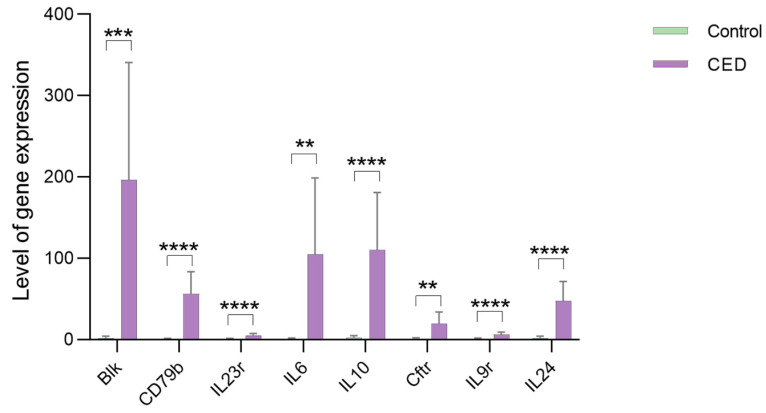
qPCR validation. The mRNA expression levels of Blk, Cftr, Cd79b, Il10, Il9r, Il24, Il23r, and Il6 were significantly higher in the CED group. ** *p* value < 0.01, *** *p* value < 0.001, **** *p* value < 0.0001.

**Table 1 cimb-47-00670-t001:** CED dataset information list.

	**CED Dataset**
Species	Rattus norvegicus
Tissue	Corneal endothelium
Samples in CED group	4
Samples in control group	4

**Table 2 cimb-47-00670-t002:** GSVA enrichment analysis results.

Description	logFC	*p* Adjust
Palomero gsi sensitivity dn	1.59117	0.0000691
Reactome propionyl coa catabolism	1.34145	0.000139
Bruneau septation atrial	1.28909	0.000257
Kaposi liver cancer met dn	1.27419	0.000116
Wp fatty acid omegaoxidation	1.22521	0.0000691
Wang THOC1 targets up	1.22475	0.000187
Wp aflatoxin B1 metabolism	1.21478	0.02236
Reactome regulation of gene expression in endocrine commited neurog3 progenitor cells	1.21111	0.000192
Smid breast cancer relapse in pleua up	1.20045	0.000073
Reactome SUMO is proteolytically processed	1.19211	0.000116
Reactome FLT3signaling through Src family kinases	1.306768	0.000073
Reactome alternative complement activation	1.31033	0.000069
Wilson Proteases at tumor bone interface dn	1.312847	0.0000691
Aujla IL22 and IL17Asignaling	1.322637	0.0000691
Reactome intestinal absorption	1.354629	0.000131
Biocarta neutrophil pathway	1.355318	0.0000691
Reactome metal sequestration by antimicrobial proteins	1.376517	0.0000855
Cheng TAF7L targets	1.413491	0.0000782
Myllkangas amplification hot spot 1	1.41585	0.0000691
Reactome alpha defensins	1.495057	0.0000691

## Data Availability

All relevant data are within the manuscript and its additional files. All data used in this work can be acquired from the GeneCards database (https://www.genecards.org/ (accessed on 12 April 2023)), MSigDB database (http://doi.org/10.1016/j.cels.2015.12.004), STRING database (http://doi.org/10.1093/nar/gky1131), Alphafold website (https://www.alphafold.ebi.ac.uk/ (accessed on8 April 2023)), miRDB database (http://doi.org/10.1093/nar/gkz757), ChIPBase database (http://doi.org/10.1093/nar/gkw965 (accessed on 5 May 2023)), and ENCORI [31] database (https://starbase.sysu.edu.cn/ (accessed on 15 May 2023)).

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
