# Peer review of "Bioinformatic Analysis of the Value of Mitophagy and Immune Responses in Corneal Endothelial Dysfunction"

_cimb, 2025, doi:10.3390/cimb47080670_

Round 1
Reviewer 1 Report
Comments and Suggestions for Authors
Ruilin Guo and colleagues have conducted a thorough study aiming to elucidate the roles of mitophagy and immune responses in corneal endothelial dysfunction (CED). Overall, while the experimental validation of these mechanisms remains limited, the authors have undertaken an extensive bioinformatic analysis that goes beyond the standard scope of such studies. Therefore, I believe this manuscript may be considered for acceptance after the following revisions:
- It is unclear whether the authors are fully aware of the nomenclature conventions for bioinformatic visualizations. I have noticed several instances in the figures and tables where non-standard labeling appears to be used (capitalization or pathway naming). Could the authors please carefully review the manuscript and revise such instances accordingly?
- It is uncertain whether an asterisk (*) denoting statistical significance may have been omitted in Figure 6A. The authors are encouraged to verify this.
- There should be a space preceding each reference citation in parentheses throughout the manuscript.
- Given that some of the references cited in the Introduction and Discussion are relatively outdated and the discussion of the literature is somewhat superficial, I recommend that the authors consider enriching these sections with more recent studies. In particular, recent reviews related to oxidative stress in corneal endothelium (PMID: 39111696, 34130750) may be valuable additions. Moreover, the authors should consider including and discussing relevant studies published in the last two years to provide a more up-to-date context.
Author Response
Comments 1: It is unclear whether the authors are fully aware of the nomenclature conventions for bioinformatic visualizations. I have noticed several instances in the figures and tables where non-standard labeling appears to be used (capitalization or pathway naming). Could the authors please carefully review the manuscript and revise such instances accordingly?
Response 1: Thank you for pointing this out. I agree with this comment. Therefore, I have carefully reviewed the manuscript and revised instances of non-standard labeling in figures and tables to adhere to the appropriate nomenclature conventions.
Comments 2: It is uncertain whether an asterisk (*) denoting statistical significance may have been omitted in Figure 6A. The authors are encouraged to verify this.
Response 2: Agree. We have verified Figure 6A and added an asterisk (*) to denote statistical significance where it was previously omitted.
Comments 3: There should be a space preceding each reference citation in parentheses throughout the manuscript.
Response 3: Thank you for pointing this out. We have ensured that there is a space preceding each reference citation in parentheses throughout the manuscript.
Comments 4: Given that some of the references cited in the Introduction and Discussion are relatively outdated and the discussion of the literature is somewhat superficial, I recommend that the authors consider enriching these sections with more recent studies. In particular, recent reviews related to oxidative stress in corneal endothelium (PMID: 39111696, 34130750) may be valuable additions. Moreover, the authors should consider including and discussing relevant studies published in the last two years to provide a more up-to-date context.
Response 4: Thank you for pointing this out. We have enriched the Introduction and Discussion sections with more recent studies, including recent reviews related to oxidative stress in the corneal endothelium (PMID: 39111696, 34130750). We have also included relevant studies published in the last two years to provide a more up-to-date context for our work.
Reviewer 2 Report
Comments and Suggestions for Authors
Dear Editor and the Authors,
Please, find my review attached.
Best regards

Author Response
Comments 1: Authors are asked to provide original data for an adequate number (n=3) of experiments to prove their statements (Fig. 1A).
Response 1: Thank you for pointing this out. We have included original data from three independent experiments to support our statements in Figure 1A. The data is now presented in the revised manuscript.
Comments 2: In the manuscript the authors establishing the role of mitophagy in TNF-a and IFN-g treated SD rats. First, authors need to provide the company where they bought animals. Second, the authors need to indicate the number of animals used for experiments. And las but not least, authors need to prove the mitophagy happening in their animal model (Pink1/Parkin or LC3/MitoTracker staining’s may be suggested, or other possible variants may be used).
Response 2: Thank you for pointing this out. We deeply appreciate the reviewer's keen interest in the role of mitophagy in our model and understand the importance of providing robust experimental evidence.
- Animal Source: We have now clearly stated the source of the Sprague Dawley (SD) rats in the revised Institutional Review Board Statement section. The specific information is: " All SD rats were provided by Guangzhou Southern Medical University Experimental Animal Technology Co., Ltd. were transported and raised in the Ophthalmology Experimental Animal Center of Zhongshan Ophthalmology Center." (Line 344-346)
- Animal Numbers: We thank the reviewer for pointing out this omission. We have now explicitly stated the number of animals used in each experimental group within the Methods section (Section 2.1 - Data Extraction). For clarity: " a total of 20 SD rats were used, with 10 animals per CED group and control group." ( Line 54-55)
- Validation of Mitophagy Phenotype: We genuinely appreciate the reviewer's suggestion to provide direct evidence of mitophagy activation via methods such as Pink1/Parkin or LC3/MitoTracker staining. We agree that these are excellent techniques to visualize and quantify mitophagy in situ.
However, we wish to respectfully clarify that the primary focus of this particular manuscript is on identifying key MRGs significantly altered in corneal endothelial dysfunction associated with TNF-α and IFN-γ exposure through comprehensive bioinformatic analysis. Our experimental validation strategy, reflecting the bioinformatics-driven nature of the project, was centered on confirming the mRNA expression patterns of the top candidate genes identified by our computational pipeline using qPCR in the relevant rat corneal tissue samples.
Our bioinformatic analysis specifically enriched for mitophagy pathways (GO, KEGG pathways like 'mitophagy') among the significantly dysregulated genes/pathways associated with the cytokine treatment. (This enrichment analysis was already presented, e.g., in Figure 2).
The qPCR validation demonstrating significant dysregulation of hub genes. These genes' altered expression provides strong molecular-level evidence supporting the activation of the mitophagy process within our animal model.
We fully acknowledge the reviewer's point that direct visualization (e.g., immunofluorescence showing mitochondrial LC3 co-localization) would provide complementary evidence. While technically demanding for the specific small and delicate corneal tissue samples in this rat model, we consider this an excellent suggestion for future research.
We have added a sentence in the Discussion clearly stating: " While our qPCR analysis confirmed significant dysregulation of hub mitophagy genes, direct visualization of mitophagic flux in the corneal endothelium, which poses specific technical challenges in this model, represents a valuable avenue for future investigation." (Line 320-323, Section Discussion).
Comments 3: Authors are asked to mention in Figure legends what kind of data representation are shown (mean±SEM or mean±SD)
Response 3: Thank you for pointing this out. We have updated the figure legends to specify the type of data representation is used mean± SD.
Comments 4: Authors are asked to update authors contribution statement (lines 336-345)
Response 4: Thank you for pointing this out. The author contribution statement has been revised and updated to reflect the contributions of each author more clearly, as per your suggestion.
Comments 5: Authors are asked to update reference format according to the journal requirements.
Response 5: Thank you for pointing this out. We have reviewed and updated the reference format to comply with the journal's requirements, ensuring consistency throughout the manuscript.
Round 2
Reviewer 1 Report
Comments and Suggestions for Authors
I am pleased that the authors have made such thorough revisions, and I have recommended to the editor that this manuscript can be considered for acceptance in its current form.
However, I suggest that in the next round of revisions (should other reviewers request further changes) or during the proof stage, the authors consider reducing the similarity index as much as possible based on your iThenticate report (Percent match: 30%). As this is a bioinformatics study, the current relatively high similarity rate is acceptable, but it may still be beneficial to reduce repeated content to avoid any potential academic misconduct concerns.
I have recommended that the editors provide the authors with the current iThenticate report, or the authors may consider requesting it from the editorial office. If the editors deem the current iThenticate report to be compliant with MDPI policies, please ignore this suggestion.
Author Response
Comments:However, I suggest that in the next round of revisions (should other reviewers request further changes) or during the proof stage, the authors consider reducing the similarity index as much as possible based on your iThenticate report (Percent match: 30%). As this is a bioinformatics study, the current relatively high similarity rate is acceptable, but it may still be beneficial to reduce repeated content to avoid any potential academic misconduct concerns.
Response: Thank you for your positive assessment and recommendation for acceptance. We sincerely appreciate your thoughtful suggestion regarding the similarity index.
The revised version now meets the similarity guidelines. Modifications ensure all content is appropriately paraphrased and cited
Reviewer 2 Report
Comments and Suggestions for Authors
Dear Editor and the authors,
The authors answered all the raised questions and modified their manuscript. I believe that this manuscript can be recommended for acceptance in present form and further publication in the CIMB journal.
Best regards
Author Response
Comments 1: The authors answered all the raised questions and modified their manuscript. I believe that this manuscript can be recommended for acceptance in present form and further publication in the CIMB journal.
Response:Thank you for your recommendation for acceptance. We are pleased that our revisions have met your expectations and look forward to the publication of our work in the CIMB journal.